

# Renewable Energy Complementarity (RECom) Maps - a comprehensive visualisation tool to support spatial diversification

Til Kristian Vrana[1,2,*] and Harald G. Svendsen[1,*]

[1]SINTEF Energi, Trondheim, Norway
[2]Equinor, Trondheim, Norway
[*]These authors contributed equally to this work.

**Correspondence:** Til Kristian Vrana (vrana@alumni.ntnu.no)

**Abstract.** Maps showing the mean wind speed only give an inaccurate indication on the quality of locations for future wind power developments. Calculating the capacity factor (based on a weather time series and a wind turbine power curve) and plotting that on a map gives a better indication on the expected mean power output, but the outcome depends on the turbine choice. In this article, the mentioned dependency is compensated for by putting the expected mean power output in relation

to the expected mean power output of all other wind parks of the region (assuming that they use the same turbine type). This *relative capacity factor* results in nice comprehensive wind resource maps, and can be plotted for the situation today and also for a future scenario. Since the expected income of a potential wind park is the product of mean power output and mean market value, looking at the relative capacity factor only does not give the full picture. The mean market value is influenced by the merit order effect that is mainly driven by covariance with other wind parks and the capacity factor's relation to production at

low-wind moments. A *market value factor* is introduced that captures the expected mean market value relative to other wind parks, based on a simplified power market model. Finally the *RECom index* is defined, combining the relative capacity factor and market value factor into a single index, resulting in the RECom map. This map can comprehensively show the revenue potential of different locations for potential future wind power developments.

## 1 Introduction

The Energiewende, with its shift from fossil-fuel-based electric power sources to weather-driven sources, leads to increasing total variance of the generation fleet's power output (or more correctly the available power output). This variance is problematic for the power system, as it makes the mandatory match between generation and load more complicated Hodge et al. (2020). It is advantageous from a power system balancing perspective if total variance can be kept low, as higher variance comes along with higher balancing cost.

Spatial diversification is gaining increasing importance for the siting of renewable energy sources. It is an effective countermeasure to limit the increasing variance of the power output from weather-driven renewable energy sources such as wind power Vrana et al. (2023). It can be imagined as a geographical low pass filter on the wind resource fluctuations, limiting the impact of local weather phenomena, such as a storm.



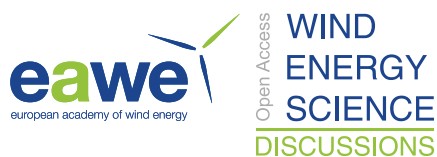

The European Green Deal and the European Commission's dedicated Offshore Renewable Energy Strategy envision more
than 300 GW of offshore wind parks in Europe European Commission (2020-11-19). A large share of it is co-located in a
rather small area: the southern North Sea. The offshore wind power developments in Europe are an example of poor spatial
diversification. Such a geographical concentration increases total variance of the wind power generation fleet, with all the
negative side effects.

When looking further into the future, it will become important to spread wind power development better in space. To
support this needed spatial diversification, a new conprehesive visualisation tool has been developed: the Renewable Energy
Complementarity maps (RECom maps). The tool is described in this article.

The RECom map combines information about the energy resource at a given location and the expected market value of
power capacity installed in this location. Somewhat similar approaches exist that map wind power according to estimated
levelised cost of energy Martinez and Iglesias (2022) Sørensen et al. (2021). Although highly relevant, such mapping does not
capture the fact that wind power impacts the power prices, generally reducing its market value with growing wind power shares
Hirth (2013). An interesting example of mapping based on market value has been made for PV in Switzerland Dujardin et al.
(2022), but there the focus is on altitude effects, not geographical distribution.

We show the RECom map on the example of wind power, but the methodology is not wind-power-specific and it can likely
be used for solar power, or for mixes of different weather-driven renewable energy sources.

## 2 Data and assumptions

This chapter describes the data used for estimating wind power resources in the North Sea and the assumptions regarding future
deployment of offshore wind parks.

### 2.1 Wind speed time series

To estimate future wind power time-series for arbitrary locations in the North Sea area, we use numerical weather model data
for historical years. This is a common approach. In our case, we have obtained wind speed data at 100 m height for the five year
period from 2018 to 2022 from the MERRA 2 dataset Molod et al. (2015), using the convenient *Renewables ninja* Renewables
Ninja (2023) interface.

### 2.2 Wind power

Wind speed time-series have been converted to wind power per installed capacity by applying a power curve $\Gamma$ representative
of a large wind park:

$$p_i(t) = \Gamma[v_i(t)] \tag{1}$$





This power curve has been obtained by a simple Gaussian filter with standard deviation $\sigma = 0.2$ applied on a single wind turbine power curve Staffell and Green (2014) Staffell and Pfenninger (2016), see Figure 1. This is done automatically by Renewables ninja.

The Gaussian filtering method to obtain wind park power curves is a simplification that does not include wake effects, and the estimated capacity factors are therefore high. However, for the present study we consider this approach sufficient, especially since this upward bias is compensated for later on (see Section 3.3), and because we are considering a future wind power scenario where wake losses is only one of many uncertainties.

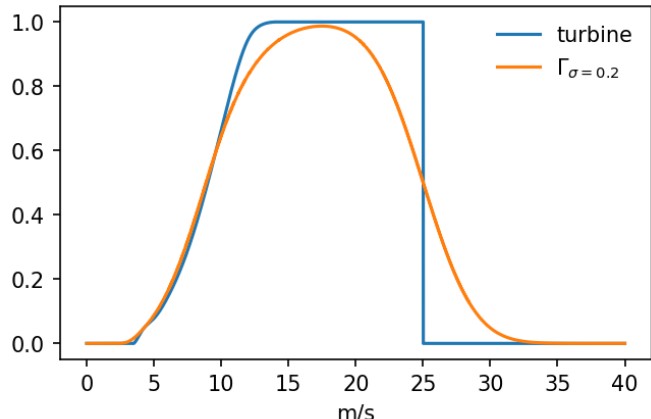

**Figure 1.** Wind park power curve $\Gamma$

## 2.3 Geographical resolution

The wind speed and wind power time-series were generated for a latitude-longitude lattice at 1.0 degree intervals in the latitude direction and 1.5 degree intervals in the longitude direction. This is indicated by the black dots in Figure 2, using a Lambert equal area projection. Boxes around these points have been defined for grouping wind power capacity. Only offshore wind power within these boxes are considered in the present study. Subsequent analyses are all based on this geographical resolution. It is clear from Figure 2 that the resolution is crude especially in the coastal areas where the wind changes a lot more in space

than far offshore.

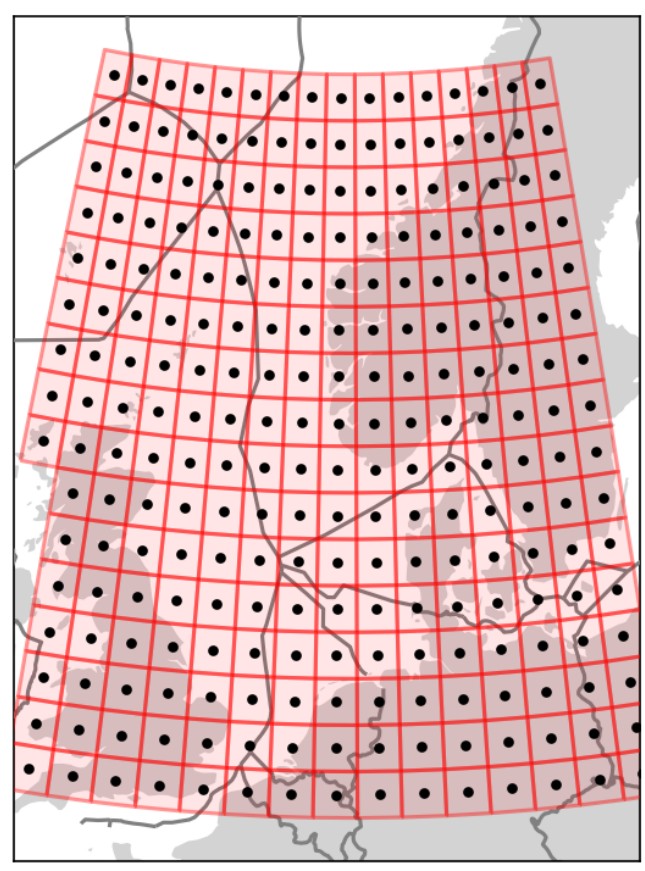

**Figure 2.** Geographical resolution

## 2.4 Wind power deployment scenarios

To plot the RECom Map, it is necessary to know the main parameters of all wind parks in the considered region. This can either be the real situation as it is today, or it can be a future scenario, as we do in this article.

Assumptions for the total offshore wind power capacity per country have been based on ENTSO-E Ten-year network development plan (TYNDP) 2022 "distributed energy" storyline assumptions ENTSO-E (2022), except for Norway, where the new target of 30 GW wind capacity by 2040 is considered instead.

The geographical distribution of the wind power capacity has been obtained using public information available via the EMODnet database European Marine Observation and Data Network (EMODnet) (2023). This database is a synthesis of national data, and includes wind park polygons representing existing and planned wind parks, and in most cases their (assumed) power capacity. Wind parks in early planning often lack information about expected power capacity.

Using these data sources, the wind deployment scenarios for 2025, 2030, 2035 and 2040 have been determined according to the following principles (see Figure 3):





1. In cases where the Emodnet data indicate too high capacity for a country, wind parks are (randomly) included until the target capacity is reached.

2. In cases where the Emodnet data indicate too low capacity for a country, the missing capacity is added to planned wind park areas that have an unspecified capacity in the dataset.

3. For 2040, some wind park locations are added manually in line with tentative plans for the various countries Arup (2022) Rijkswaterstaat (2022).

4. Finally, capacity is scaled up or down so that the sum per country matches the country target for the specific year.

5. For 2035, the average of 2030 and 2040 target capacities were assumed.

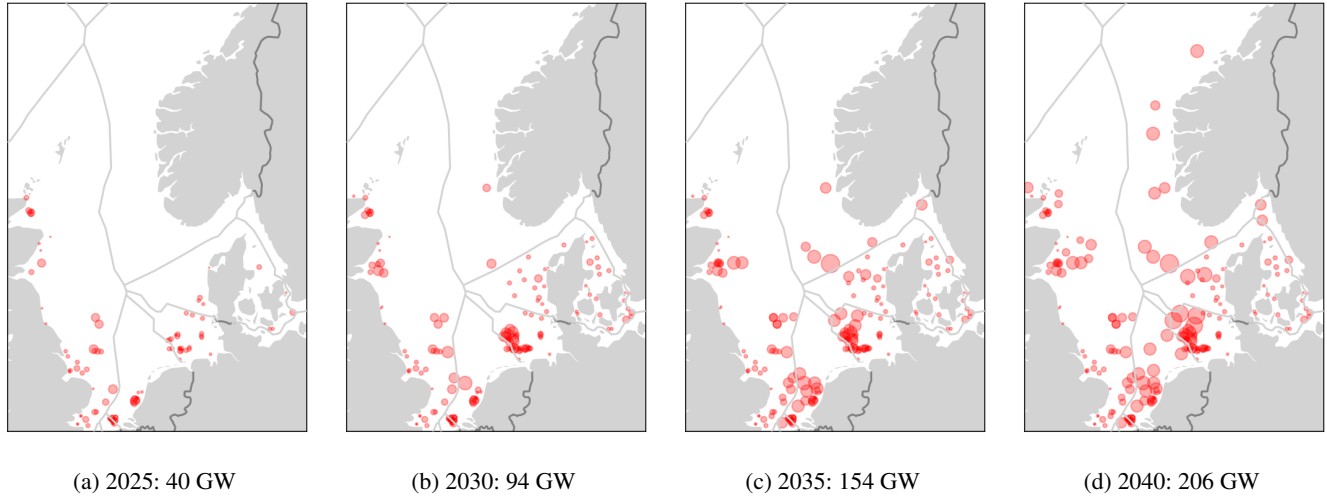

(a) 2025: 40 GW     (b) 2030: 94 GW     (c) 2035: 154 GW     (d) 2040: 206 GW

**Figure 3.** Assumed wind power capacity in the North Sea

## 3  Wind resource maps

The state of the art to assess the suitability of locations for wind power developments is to draw wind speed maps, often called a wind atlas.

### 3.1  Mean wind speed

It is common to use maps that show the mean wind speed for the goal of assessing good locations for potential new wind parks Hasager et al. (2006). Such a plot is shown in Figure 4. This map relies on data for wind speeds at a given height (Section 2.1), but nothing else.



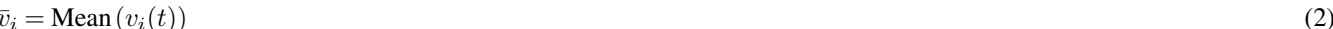

$$\bar{v}_i = \text{Mean}\,(v_i(t)) \tag{2}$$

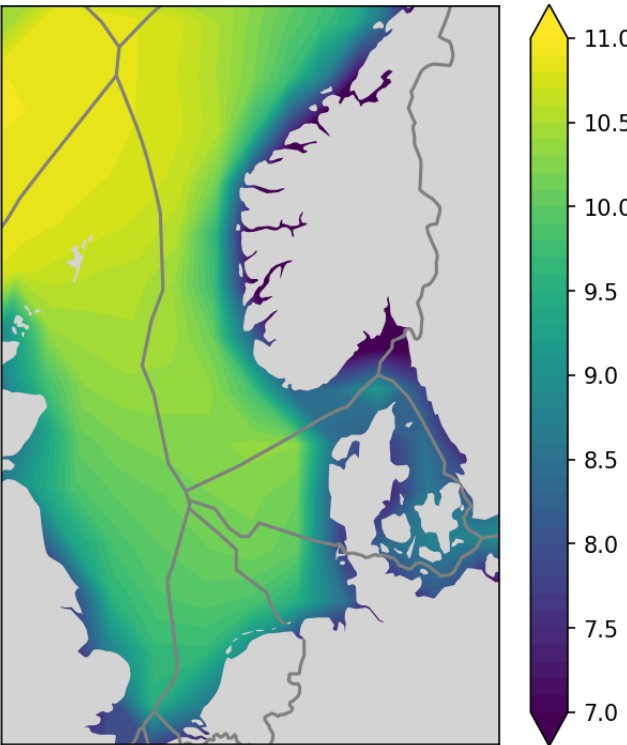

**Figure 4.** Mean wind speed at hub height [m/s]

**Room for improvement**

Mean wind speed is indeed a valid indicator for the quality of a site for wind park developments, but it does not give the full picture, as the relationship between wind power and wind speed is highly non-linear. This can be best shown with an unrealistic example:

  – site A has 6 months of 25 m/s and 6 months of 0 m/s

  – site B has 12 months of 12.5 m/s

Both sites have identical mean wind speeds, and will appear the same on such a mean wind speed map. However, the energy output of a wind park at site A will be significantly lower than for site B. This phenomenon is not accounted for when using mean wind speed maps. It is therefore better to plot capacity factor maps, as explained in the following section.



## 3.2 Capacity factor

A better indicator of wind power potential is the expected capacity factor $\bar{p}_i$, i.e. the mean power output per installed capacity:

$$\bar{p}_i = \text{Mean}(p_i(t)) = \text{Mean}\left(\Gamma[v_i(t)]\right) \tag{3}$$

In addition to data for wind speeds, this step requires assumptions about the power curve (Section 2.2). It gives a better indication on the quality of sites for potential wind parks than the mean wind speed.

**Room for improvement**

This is already an improvement compared to the mean wind speed map, as it gives a more accurate indication on the expected energy output of a wind park. However, there is still a problem: the result depends on the used power curve $\Gamma(v)$. It is not clear if a capacity factor calculated to a specific value (e.g. 45 %) indicates a good site or a bad site.

– if a power curve with large rotor was used (low specific power), 45 % might not be very good

– for another power curve with a smaller rotor (high specific power), 45 % might actually be quite good

## 3.3 Relative capacity factor

To circumvent the dependency on the wind power curve choice, we define the *relative capacity factor* which compares the expected capacity factor to the mean value for the region. First, we write the expression for the normalised aggregated wind power output, which is used as the *baseline*:

$$p_{\text{total}}(t) = \frac{1}{\hat{P}_{\text{total}}} \sum_{w \in W} P_w(t) = \sum_{w \in W} \frac{\hat{P}_w}{\hat{P}_{\text{total}}} \Gamma[v_w(t)] \tag{4}$$

$W$ denotes all wind parks and $\hat{P}_{\text{total}} = \sum_{w \in W} \hat{P}_w$ is the total installed capacity, as given by the assumed deployment scenario (Section 2.4). Here we consider only offshore wind parks, but this could readily be generalised to include onshore wind parks. Now we define the relative capacity factor $\bar{p}_i^{\text{rel}}$ as the capacity factor of location $i$ divided by the mean capacity factor of all considered wind parks:

$$\bar{p}_i^{\text{rel}} = \frac{\bar{p}_i}{\bar{p}_{\text{total}}} = \frac{\text{Mean}(p_i(t))}{\text{Mean}(p_{\text{total}}(t))} \approx 1 \tag{5}$$

The relative capacity factor $\bar{p}_i^{\text{rel}}$ has a baseline value of one. It is a meaningful quantity to plot on a map (see Figure 5) as a measure of the wind resource, where the outcome is less dependent on the specific choice of the wind turbine, the spacing between wind turbines, details on wake consideration, etc. It simply shows how the wind resource is distributed with a neutral index that is on average equal to one.

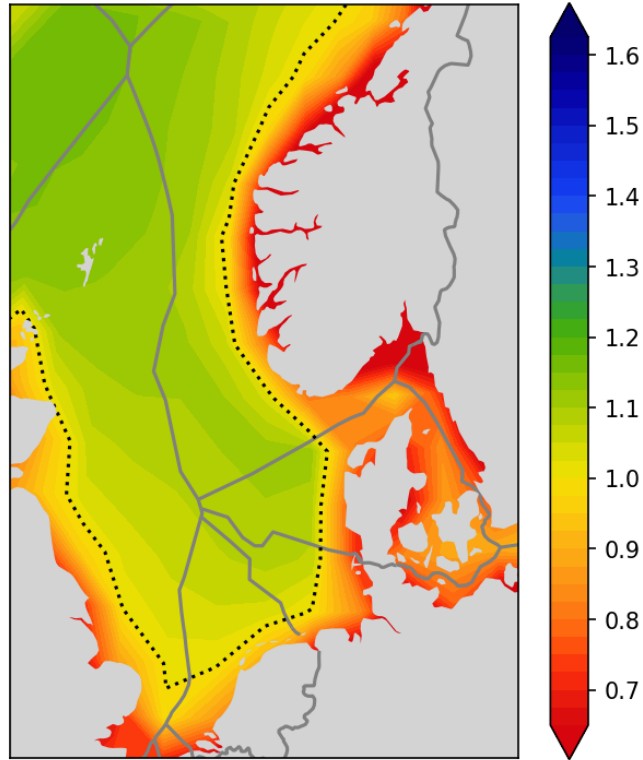

**Figure 5.** Relative capacity factor $\bar{p}_i^{\text{rel}}$ (2040) – The dotted line indicates a baseline value of one

**Room for improvement**

Even though the shown wind power map nicely indicates how much power can be expected to be harvested at given locations, it
lacks an indication of the revenue that can be expected from that generated power. Wind power that is generated simultaneously
with many other wind parks will be valued less on the electricity market. It is therefore not enough to only look at the wind
resource for identifying good locations for potential new wind parks. It is important to also look at the placement of other wind
parks and the *covariance* between wind parks.

## 4   Covariance of wind power

Even though the relative capacity factor map provides a nice overview on which location will likely provide a good power
output, it does not say anything about which location will provide good revenue on the electricity market.

Due to the merit order effect Sensfuß et al. (2008) Antweiler and Muesgens (2021) Ketterer (2014), it can be expected that
wind power output, that comes at time when many other wind parks provide high output as well, will achieve low value on the





market. It can also be expected that wind power output that comes at times when the other wind parks cannot produce electric
power will be valued higher on the market. A suitable mathematical concept to assess these phenomena is the *covariance*.

## 4.1   Covariance

The covariance between two time series $X$ and $Y$ is defined as the mean value of the product of their deviation from their
respective mean values, i.e.

$$\mathrm{Cov}(X,Y) = \mathrm{Mean}\left((X(t) - \bar{X})(Y(t) - \bar{Y})\right). \tag{6}$$

The *variance* of a time series $X(t)$ is its covariance with itself, and the *standard deviation* $\sigma_X$ is the square root of the
*variance*, i.e.

$$\sigma_X^2 = \mathrm{Var}(X) = \mathrm{Cov}(X,X). \tag{7}$$

The *correlation coefficient* is the covariance normalised by the standard deviations and will always have values in the interval
from $-1$ to $1$:

$$\mathrm{Corr}(X,Y) = \frac{\mathrm{Cov}(X,Y)}{\sigma_X \sigma_Y} \tag{8}$$

For the variance of a sum of independent variables, the following general expression holds:

$$\mathrm{Var}(X + Y) = \mathrm{Var}(X) + \mathrm{Var}(Y) + 2\mathrm{Cov}(X,Y). \tag{9}$$

Covariance is a mathematical operation that is very useful for spatial planning of renewable energy. Often *correlation
coefficients* are discussed for such purposes, but that has disadvantages as they only indicate the similarity of the fluctuations
of the weather between two locations and no information about the amplitude of the fluctuations. Covariance includes both and
is therefore more appropriate for assessing potential sites.

Example: Two nuclear power stations give an almost flat output profile, with tiny but identical fluctuations. Is that correlated?
Yes. Is it a problem? no. The tiny fluctuations might be highly correlated, but that does not matter, because they are so small.
In this case correlation is high, but covariance is low.

## 4.2   Covariance of normalised power time series

Let us introduce some simplifying notation. We express power time series in terms of capacity and normalised power output,
$P_x(t) = \hat{P}_x \cdot p_x(t)$. Furthermore, we write

$$\sigma\left(p_x(t)\right) = \sigma_x \tag{10}$$

$$\mathrm{Var}\left(p_x(t)\right) = \sigma_x^2 = \mathrm{var}_x \tag{11}$$

$$\mathrm{Corr}\left(p_x(t), p_y(t)\right) = \mathrm{corr}_{x,y} \tag{12}$$

$$\mathrm{Cov}\left(p_x(t), p_y(t)\right) = \mathrm{cov}_{x,y} \tag{13}$$



### 4.3 Relative covariance

For a candidate new wind park (indicated with the index "$i$") and the entire wind power fleet (indicated with the index "total"), we define the relative standard deviation $\sigma_i^{\mathrm{rel}}$ with a baseline value of one, similar to the relative capacity factor $\bar{p}_i^{\mathrm{rel}}$ defined in Equation (5):

$$\sigma_i^{\mathrm{rel}} = \frac{\sigma_i}{\sigma_{\mathrm{total}}} \approx 1 \tag{14}$$

We also define the relative covariance $\mathrm{cov}_i^{\mathrm{rel}}$ with a baseline value of one:

$$\mathrm{cov}_i^{\mathrm{rel}} = \frac{\mathrm{cov}_{i,\mathrm{total}}}{\mathrm{var}_{\mathrm{total}}} = \sigma_i^{\mathrm{rel}} \cdot \mathrm{corr}_{i,\mathrm{total}} \tag{15}$$

The relative covariance consists of two terms. The first term, $\sigma_i^{\mathrm{rel}}$, is generally larger than zero, because aggregation effects lead to a generally lower variability of the entire wind power fleet as compared to individual wind parks. However, there is no mathematical necessity than it always needs to be larger than one. At sites with very low wind speeds, values lower than one occur. The second term, $\mathrm{corr}_{i,\mathrm{total}}$, is by definition limited to be at most equal to one.

The product of these two terms, $\mathrm{cov}_i^{\mathrm{rel}}$ can therefore become both larger and smaller than one. It gives an indication on the quality of a site for potential wind parks with regards to the covariance of the wind power output of that site with the output of the complete wind power fleet.

The relative covariance can be interpreted as the ratio of how much the overall variance increases when adding a new wind park of infinitesimal capacity at a specific location $i$ compared to scaling up total capacity by the same amount. To see this, consider an infinitesimal capacity increase $\delta p$, and the resulting increase in overall variance, $\delta \mathcal{V}$. If we add a new wind park $i$, we get:

$$\delta \mathcal{V}_i = \mathrm{Var}(P_{\mathrm{total}} + p_i \delta p) - \mathrm{Var}(P_{\mathrm{total}}) \tag{16}$$

$$= \underbrace{(\delta p)^2 \mathrm{Var}(p_i)}_{\to 0} + 2\delta p \hat{P}_{\mathrm{total}} \mathrm{Cov}(p_i, p_{\mathrm{total}}) \tag{17}$$

Here, we have suppressed the time dependence, $p_i = p_i(t)$, $p_{\mathrm{total}} = p_{\mathrm{total}}(t)$, and $P_{\mathrm{total}} = \hat{P} p_{\mathrm{total}}(t)$, and used Equation (9).

Similarly, if we scale up existing capacity by the same amount $\delta p$, we get:

$$\delta \mathcal{V}_{\mathrm{scale}} = \mathrm{Var}(P_{\mathrm{total}} + p_{\mathrm{total}} \delta p) - \mathrm{Var}(P_{\mathrm{total}}) \tag{18}$$

$$= \underbrace{(\delta p)^2 \mathrm{Var}(p_{\mathrm{total}})}_{\to 0} + 2\delta p \hat{P}_{\mathrm{total}} \mathrm{Cov}(p_{\mathrm{total}}, p_{\mathrm{total}}) \tag{19}$$

Since $\delta p$ is infinitesimal, the $(\delta p)^2$ terms disappear and we can conclude that the ratio is indeed the relative covariance from Equation (15):

$$\frac{\delta \mathcal{V}_i}{\delta \mathcal{V}_{\mathrm{scale}}} = \frac{\mathrm{cov}_{i,\mathrm{total}}}{\mathrm{var}_{\mathrm{total}}} = \mathrm{cov}_i^{\mathrm{rel}} \tag{20}$$





Now we can summarise what different values of the relative covariance for a new wind park at location $i$ imply:

$\mathbf{cov}_i^{\mathbf{rel}} > 1$ : gives more total variance increase than a flat scale-up of the existing wind power fleet ($\delta\mathcal{V}_i > \delta\mathcal{V}_{\text{scale}}$), which is the worst case

$\mathbf{cov}_i^{\mathbf{rel}} = 1$ : gives the same total variance increase as scaling up existing wind power fleet ($\delta\mathcal{V}_i = \delta\mathcal{V}_{\text{scale}}$), which is the baseline case

$0 < \mathbf{cov}_i^{\mathbf{rel}} < 1$ : gives less total variance increase than a flat scale-up of the existing wind power fleet ($0 < \delta\mathcal{V}_i < \delta\mathcal{V}_{\text{scale}}$),
which is good and possibly the best realistic case

$\mathbf{cov}_i^{\mathbf{rel}} = 0$ : does not correlate with the existing wind power fleet and therefore does not influence total variance ($\delta\mathcal{V}_i = 0$)

$\mathbf{cov}_i^{\mathbf{rel}} < 0$ : anti-correlates with the existing wind power fleet, and reduces total variance ($\delta\mathcal{V}_i < 0$), which is the theoretical best case

### 4.4 Covariance equivalent installed capacities

Based on the relative covariance, an expression for the covariance equivalent installed capacity can be formulated:

$$\hat{P}_i^{\text{eqv}} = \hat{P}_{\text{total}} \cdot \text{cov}_i^{\text{rel}} \tag{21}$$

The covariance equivalent installed capacity has the unit W, which is suitable to be understood and communicated. On the contrary, it should be noted that the variance and covariance have the unit $\text{W}^2$, which is somewhat non-intuitive and difficult to interpret. However, it remains challlenging to understand what the covariance equivalent installed capacity exactly means. It
can be interpreted in the following way:

*The covariance equivalent installed capacity with full correlation with $i$ and the same variance as $i$ has the same covariance with $i$ as the total existing wind power fleet.*

The covariance equivalent installed capacity gives an indication on how correlated a potential wind park at location $i$ is with the wind power fleet. It shows how much wind power will fluctuate "synchronised" with the potential new wind park. Even
though complicated, it might be perceived easier to understand than covariance. It can also be used to show how crowded the southern North Sea will become in the future, see Figure 6. It is a nice way to "compress" all the individual wind parks in Figure 3, achieving a simpler graphical representation of the influence of the existing wind power fleet.

Here we show the covariance equivalent installed capacity maps for all the four scenario time steps. The plots nicely illustrate how the situation gets worse over time as more wind power capacity is added.
However, the covariance equivalent installed capacities shown do not directly indicate good or bad sites for new wind parks. A value of $50\,\text{GW}$ might be bad in 2025, but it is good in 2040, as the levels are generally much higher.





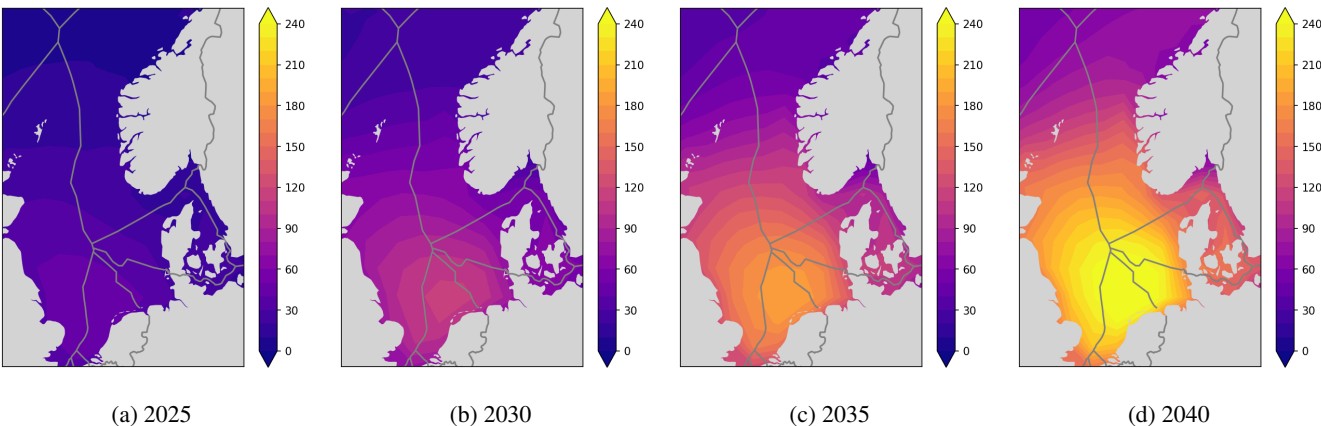

| (a) 2025 | (b) 2030 | (c) 2035 | (d) 2040 |

**Figure 6.** Covariance equivalent installed capacities $\hat{P}_i^{\text{eqv}}$

### 4.5 Implications for wind power maps

Adding a new wind park in the yellow area of Figure 6d, that will reach high power output mostly at times when a large part of the North Sea wind power fleet is doing exactly the same, might not be a good idea. Even if the wind resources in that area is good, as shown in Figure 5, the produced power will likely achieve a reduced value on the electricity market. It is therefore not sufficient to look at relative capacity factors as in Figure 5 for determining good locations for future wind power developments. Another type of map is needed that accounts for both the wind resource and the *complementarity* with other wind parks. This is introduced in the following section.

## 5 RECom maps

In this section, we develop the Renewable Energy Complementarity (RECom) map, which aims to address the drawbacks for normal wind resource maps, and which accounts for covariance with existing wind parks.

### 5.1 Complementarity factor

Now, based on the relative covariance $\text{cov}_i^{\text{rel}}$ in Equation (15), the complementarity factor is defined:

$$\Phi_i = 1 + \beta \left( 1 - \underbrace{\text{cov}_i^{\text{rel}}}_{\approx 1} \right) \approx 1 \tag{22}$$

It has a baseline value of one and contains the same information as the relative covariance $\text{cov}_i^{\text{rel}}$, but it is brought to a format which corresponds well with the relative capacity factor $\bar{p}_i^{\text{rel}}$ from Equation (5) (shown in Figure 5), as it shares the following properties:





- higher values indicate better sites

- average sites score a value of one

The parameter $\beta$ is a positive number and represents the weighting of how much higher $\Phi_i$ an uncorrelated production profile achieves compared to a fully correlated production profile. There is no correct quantification of $\beta$, it is simply is a choice. We chose to operate with $\beta = 0.5$. The sensitivity towards the choice of $\beta$ is elaborated in Section 6.4.

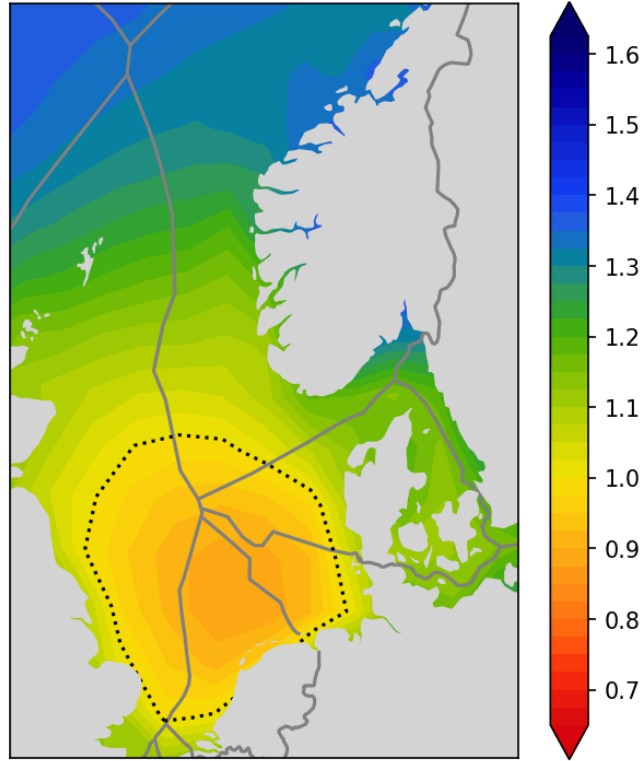

**Figure 7.** Complementarity factor $\Phi_i$ (2040) ($\beta = 0.5$)

The complementarity factor $\Phi_i$ gives an indication on the *value* of wind power, as it accounts for covariance and the fact that uncorrelated wind power is worth more than correlated wind power. It does however not give the full picture regarding the
value of wind power. One central aspect is missing: the fact that higher capacity factors result in higher average wind power value, as the additional energy capture (compared to a site with low capacity factor) happens during *low wind hours*. This additional production is therefore worth more on average than the baseline production during high wind hours. To capture this relevant phenomenon, the *market value factor* is introduced.





## 5.2 Market value factor

Let us try to define a quantity that is an indicator of market value. The obtained income from a wind park depends on the capacity factor and the market price. Wind power has low marginal costs, so more wind power generally means lower prices. For simplicity, let us consider the price $y$ to be a linear function of the total wind power output $p_{\text{total}}(t)$ with a negative slope:

$$y(t) = \bar{y} - \frac{\alpha\bar{y}}{\bar{p}_{\text{total}}}\left(p_{\text{total}}(t) - \bar{p}_{\text{total}}\right) \tag{23}$$

Here, a bar denotes mean value. The parameters have been chosen such that at mean wind power, $p_{\text{total}}(t) = \bar{p}_{\text{total}}$, we get

$y(t) = \bar{y}$. And if there is no wind power, $p_{\text{total}}(t) = 0$, we get $y(t) = \bar{y}(1 + \alpha)$. So the $\alpha$ parameter indicates how much higher the price is when there is no wind compared to mean wind.

With this assumption, we can compute an expression for the capture price, i.e. the mean value of the electricity price weighted according to energy production. For wind park $i$, this is:

$$V_i = \frac{\sum_i \left(\hat{P}_i p_i \cdot y\right)}{\sum_i \left(\hat{P}_i p_i\right)} = \frac{1}{\bar{p}_i}\text{Mean}\left(p_i y\right)$$

$$= -\frac{\alpha\bar{y}}{\bar{p}_i \bar{p}_{\text{total}}}\text{Mean}\left((p_i - \bar{p}_i)(p_{\text{total}} - \bar{p}_{\text{total}})\right) + \bar{y}$$

$$= \left(1 - \alpha\frac{\sigma_{\text{total}}}{\bar{p}_{\text{total}}}\frac{\sigma_i}{\bar{p}_i}\text{corr}_{i,\text{total}}\right)\bar{y} \tag{24}$$

where we have simplified the notation by omitting the time dependence, $p_i = p_i(t)$, $p_{\text{total}} = p_{\text{total}}(t)$, and $y = y(t)$. For the total wind power fleet, we similarly get

$$V_{\text{total}} = \frac{1}{\bar{p}_{\text{total}}}\text{Mean}\left(p_{\text{total}}y\right) = \left(1 - \alpha\frac{\sigma_{\text{total}}^2}{\bar{p}_{\text{total}}^2}\right)\bar{y} \tag{25}$$

We define the *market value factor* $\Psi_i^{\text{lin}}$ as the ratio of these:

$$\Psi_i^{\text{lin}} = \frac{V_i}{V_{\text{total}}} = 1 + \beta\left(1 - \underbrace{\frac{\text{cov}_i^{\text{rel}}}{\bar{p}_i^{\text{rel}}}}_{\approx 1}\right) \approx 1 \tag{26}$$

Here, $\bar{p}_i^{\text{rel}} = \bar{p}_i/\bar{p}_{\text{total}}$, and $\beta$ is a positive number defined through $\frac{1}{1+\beta} = 1 - \alpha\frac{\sigma_{\text{total}}^2}{\bar{p}_{\text{total}}^2}$, and is a number given by the characteristics of the power market and the total wind power fleet.

The parameter $\beta$ represents the expected value increase of an uncorrelated production profile compared to a fully correlated

production profile. It corresponds nicely with the parameter $\beta$ from the complementarity factor $\Phi_i$ in Equation (22) in the previous subsection.

The market value factor $\Psi_i^{\text{lin}}$ has a baseline value of one. It is similar to the complementarity factor $\Phi_i$, but with the difference that the factor $\text{cov}_i^{\text{rel}}$ is replaced by $\text{cov}_i^{\text{rel}}/\bar{p}_i^{\text{rel}}$, a modification that precisely addresses the missing aspect of $\Phi_i$ identified above.

For the data we use in this article, $\sigma_{\text{total}} = 0.22$, $\bar{p}_{\text{total}} = 0.53$, and a value $\beta = 0.5$ therefore corresponds to $\alpha = 1.9$.





Remember that $\mathrm{cov}_i^{\mathrm{rel}} = \mathrm{corr}_{i,\mathrm{total}} \cdot \sigma_i^{\mathrm{rel}}$. From Equation (26), it can be seen that if $\mathrm{corr}_{i,\mathrm{total}} = 0$, then $\Psi_i^{\mathrm{lin}} = 1 + \beta > 1$. If $\mathrm{corr}_{i,\mathrm{total}} > 0$, then $\Psi_i^{\mathrm{lin}}$ decreases for larger values of $\sigma_i^{\mathrm{rel}}$. This is natural, as more variability correlated with existing variability is bad. In the case of negative correlation $\mathrm{corr}_{i,\mathrm{total}} < 0$, $\Psi_i^{\mathrm{lin}}$ *increases* for larger values of $\sigma_i^{\mathrm{rel}}$. In that case, larger variance is improving the complementary. For a "typical" wind park location, $\Psi_i^{\mathrm{lin}} \approx 1$, and for bad locations, it will be less than one, and for good locations, it will be larger than one.

### 5.3   Exponential extrapolation

To represent the relation between wind power output and electricity market revenue, a simplified linear relation was used in Equation (23). This linear assumption is advantageous, as it enables for deriving comprehensive equations such as Equation (26). But in contrary to the nice microscopic characteristics, the linear assumption does not have acceptable macroscopic characteristics.

The relative covariance $\mathrm{cov}_i^{\mathrm{rel}}$ can in theory be any real number, leading to $\Psi_i^{\mathrm{lin}}$ potentially being any real number. Negative values of $\Psi_i^{\mathrm{lin}}$, however, no not make sense as the value of the ability to havest energy from the wind cannot become negative. In the worst case, the value can converge towards zero, if there is an abundance of available power in the system, resulting in high levels of curtailment.

    To assure that the market value factor $\Psi_i^{\mathrm{lin}}$ remains positive for all possible input, it is therefore necessary to select another
function (other than linear) with acceptable global behaviour. This function would need the following properties:

– converges towards zero for large positive input

– diverges towards infinity for large negative input

– behaves similar to the previously used linear function in the realistic range for wind power data

    Recall that the Taylor expansion of the exponential function is

$$e^{\beta(1-x)} = 1 + \beta(1-x) + \frac{\beta^2}{2}(1-x)^2 + \dots \tag{27}$$

Equation (26) is therefore the first order Taylor expansion of the exponential function $e^{\beta(1-x)}$ for $x = \mathrm{cov}_i^{\mathrm{rel}} / \bar{p}_i^{\mathrm{rel}}$, and a good approximation of it if $x \approx 1$. This exponential function has the desired properties, including the same value and same derivative in the baseline point $x = 1$. We therefore introduce the exponential version of the value function:

$$\Psi_i^{\mathrm{exp}} = e^{\beta\left(1 - \frac{\mathrm{cov}_i^{\mathrm{rel}}}{\bar{p}_i^{\mathrm{rel}}}\right)} \approx 1 \tag{28}$$

It behaves similarly at the baseline point, but ensures acceptable global behaviour (staying always positive). The relation between linear function and exponential function is shown in Figure 8.



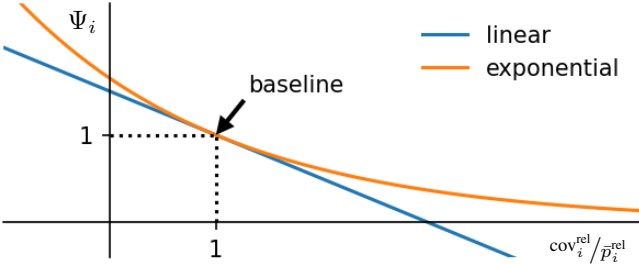

**Figure 8.** Linear vs exponential

The resulting market value factor in its exponential form is plotted in Figure 9:

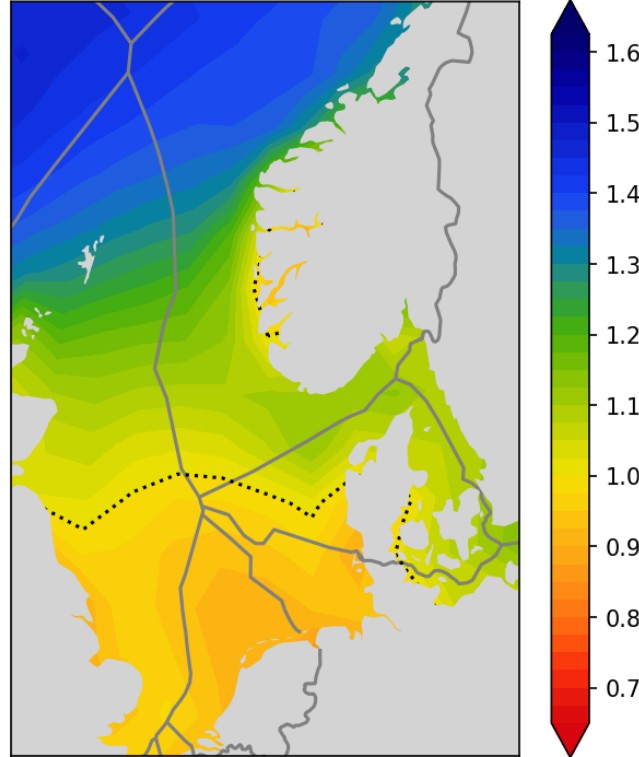

**Figure 9.** Market value factor $\Psi_i^{\text{exp}}$ (2040) ($\beta = 0.5$)

## 5.4 RECom index

Based on the relative capacity factor $\bar{p}_i^{\text{rel}}$ and the market value factor $\Psi_i^{\text{exp}}$, the RECom $\Omega_i$ index is defined as the product:





$$\Omega_i = \bar{p}_i^{\mathrm{rel}} \cdot \Psi_i^{\mathrm{exp}} = \underbrace{\bar{p}_i^{\mathrm{rel}}}_{\approx 1} \cdot \underbrace{e^{\beta\left(1 - \frac{\mathrm{cov}_i^{\mathrm{rel}}}{\bar{p}_i^{\mathrm{rel}}}\right)}}_{\approx 1} \approx 1 \qquad (29)$$

Recall that $\bar{p}_i^{\mathrm{rel}} = \bar{p}_i / \bar{p}_{\mathrm{total}}$ and $\mathrm{cov}_i^{\mathrm{rel}} = \mathrm{cov}_{i,\mathrm{total}} / \mathrm{var}_{\mathrm{total}}$.

The RECom index is plotted in Figure 10c. It gives an indication on how good a site is for wind power developments, and accounts for:

- the wind resources

– the covariance with all other wind parks

- the advantage of production at low wind hours

An average wind park gets an index value of one. Higher capacity factors result in higher index values. Less covariance with the wind power fleet will also result in higher values.

- RECom index larger than one indicates good sites

– RECom index smaller than one indicates bad sites

The RECom index is the quantity we have been seeking, and we propose to use this as a basis for maps that visualise the quality of potential wind power sites and the benefits of spatial diversification.





(a) Relative capacity factor $\bar{p}_i^{\text{rel}}$

(b) Market value factor $\Psi_i^{\text{exp}}$

(c) RECom index $\Omega_i$

**Figure 10.** Relative capacity factor, market value factor and RECom index (2040) ($\beta = 0.5$)





# 6 Sensitivities

In this section, we discuss the sensitivity of the RECom map towards several influencing parameters.

## 6.1 Linear vs. exponential

The conversion of the linear value function Equation (26) to the exponential value function Equation (28) modifies the outcome in the follow way:

– values large than one (green/blue) are amplified

– values lower than one (orange/red) are damped

The influence of the exponential formulation can be concluded from Figure 8. A comparison is shown in Figure 11.

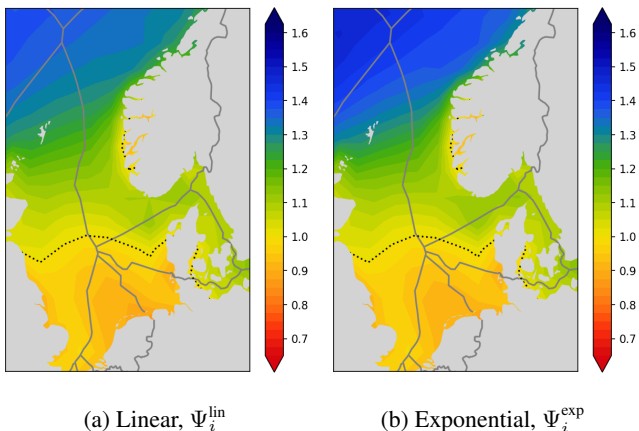

(a) Linear, $\Psi_i^{\text{lin}}$        (b) Exponential, $\Psi_i^{\text{exp}}$

**Figure 11.** Market value factor $\Psi_i$ ($\beta = 0.5$)

    As the data considered is rather close to the baseline point ($^{\text{cov}}_i^{\text{rel}}/\bar{p}_i^{\text{rel}} \approx 1$), the influence is limited. The results are therefore not distorted in an unacceptable way by the conversion. It should also be remembered that there is no reason to consider the linear formulation as *correct* reference, as the real dependency between power output and capture price is more complex than both the linear and exponential functions.

## 6.2 Sensiticity towards the scenario

The RECom index depends for a large part on the scenario and the considered wind parks. It is therefore clear that it will change over time as more and more wind parks are deployed. This development over time is shown in Figure 12.





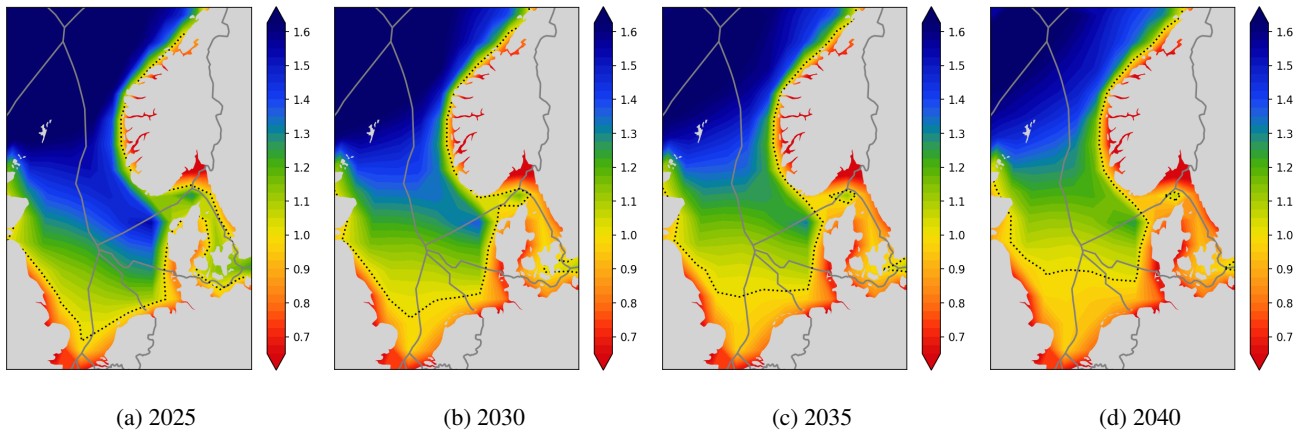

(a) 2025          (b) 2030          (c) 2035          (d) 2040

**Figure 12.** RECom index $\Omega_i$ for different years ($\beta = 0.5$)

It can be observed that the dotted line, which represents $\Omega_i = 1$ and separates good and bad locations, moves northwards with time. This is due to wind parks in the southern North Sea suffering from increasing covariance equivalent installed capacities, as

shown in Figure 6. It is, however, noticeable that the changes over time in Figure 12 are significantly smaller than in Figure 6. This is good as it gives the RECom map some level or robustness towards changes of the scenario (which will always be somewhat uncertain regarding the future). It is therefore a good measure for the suitability of sites for potential wind parks in the future.

## 6.3 Sensitivity towards the sampling rate

Hourly data was used for this study. Figure 13 shows the RECom maps of the same data after resampling to lower resolutions.

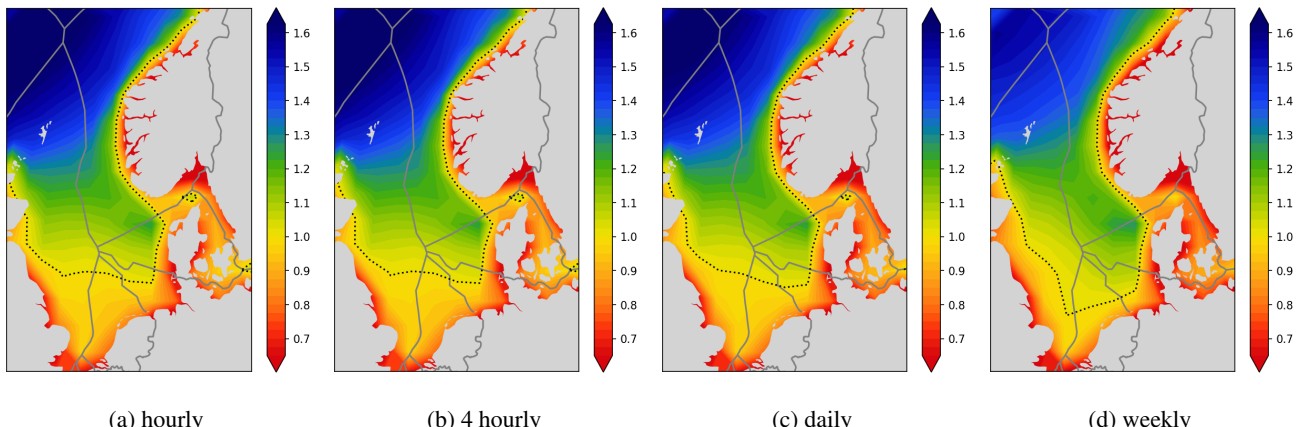

(a) hourly          (b) 4 hourly          (c) daily          (d) weekly

**Figure 13.** RECom index 2040 based on time-series with different sample rates





Again, we observe good robustness, as the changes are limited. Only in the extreme case of converting to weekly data significant changes occur. It can be concluded that the performance of the RECom index does not highly depend on high resulation data.

### 6.4 Sensitivity towards parameter $\beta$

The most influential parameter is $\beta$ which represents the dependency of the power price on the aggregated wind power output, as explained in Section 5.2. This means that the $\beta$ parameter can be thought of as a tuning parameter that weights the importance of producing power when other wind parks don't (covariance). Figure 14 shows RECom maps with different values for $\beta$.

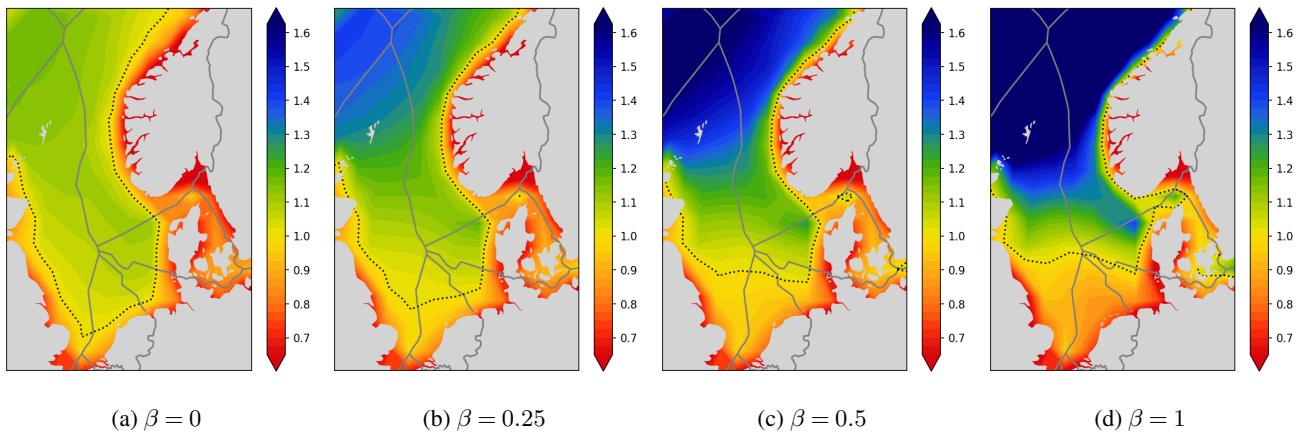

(a) $\beta = 0$    (b) $\beta = 0.25$    (c) $\beta = 0.5$    (d) $\beta = 1$

**Figure 14.** RECom index 2040 with different values of the $\beta$ parameter

Comparing the different maps shows:

- For $\beta = 0$, the RECom index becomes the same as relative capacity factor, $\Omega_i(\beta = 0) = \bar{p}_i^{\text{rel}}$, and corresponds to a
situation where the value of the wind power from a wind park is independent of wind power elsewhere, i.e. the covariance is considered irrelevant.

- For large values of $\beta$, the RECom index is dominated by the market value factor, and the $\Omega_i = 1$ line approaches the $\Psi_i^{\text{exp}} = 1$ line as seen in Figure 9 (For $\Psi_i^{\text{exp}}$ the location of this line is independent of $\beta$). This we can see already in Figure 14d. The values away from the unity line are nevertheless very different.

Our choice of $\beta = 0.5$ is somewhat arbitrary, but gives a situation where the relative capacity factor $\bar{p}_i^{\text{rel}}$ and the market value factor $\Psi_i^{\text{exp}}$ have a similar range of values around 1.0 and as such represent a similar weighing of those factors in the overall RECom map.

In reality, the most appropriate choice for $\beta$ depends on the power market situation which again depends on the scenario year and its assumptions. A better understanding of wind power capture prices in future energy systems would help choosing
a value for $\beta$. This could be obtained from detailed energy system modelling and analysis. In general, it can be stated that:





- $\beta$ increases with increasing shares of weather-driven sources in the energy system

- $\beta$ decreases with increasing load flexibility in the energy system (e.g. hydrogen electrolysis)

It is, however, not the scope of this article to perform detailed analysis of future energy system scenarios to determine $\beta$. The purpose of the RECom maps is to provide a simple way to illustrate wind power value *without* the need for energy system
modelling. Apart from the $\beta$ parameter, the maps only depend on the wind power deployment scenario without the need for any energy system modelling.

## 7    Conclusions

The RECom index and the relating RECom maps are useful tools to support spatial diversification by visualising the need and potential for it. They give a comprehensive indication on the expected revenue of potential wind park locations. Compared to
the well-known wind resource maps, that display the mean wind speed, RECom maps add significant value, as they include more highly relevant information. Their simple and comprehensive nature is intended to visualise the matter also for people who are not familiar with many of the underlaying principles, such as politicians and the general population.

The RECom index is lower in areas with a lot of wind parks, due to the high covariance and resulting lower expected wind power market value. The reality might even exaggerate this, as co-location with other wind parks might not only depress market
value due to the merit order effect, but also the power output. Areas with a lot of wind parks might be confronted with lower capacity factors (lower than expected), due to wake and blockage effects of nearby wind parks. These wave and blockage effects are, however, not yet included in the RECom index in its current form.

Even though the RECom map can show a lot more than a mean wind speed map can, one must be careful to be aware of its limitations. A few of these limitations are listed here:

- The calculation is based on a strongly simplified and idealised linear market model, which cannot fully represent the complexity of the power market price setting mechanisms.

- The choice of the parameter $\beta$ has a significant influence, while there is no *correct* way to determine it, as the future behaviour of the electric power market depends on many uncertainties.

- The electricity transmission network is not included, leaving out grid congestions and power market price zones that can
have significant influence one the revenues.

- The mean electricity price differs between countries and price zones, and the parameter $\beta$ might differ as well.

- National legislation can influence the revenue potential, but renewable energy policies and support schemes are not accounted for here.

It is obvious that a market value estimation based on a very simple model never will show the full picture. However, it is not
meaningful to get distracted by comparing the simple RECom index with a complex and time consuming power market study





of a future scenario. The RECom index and the relating RECom map need instead to be compared to mean wind speed maps, where the RECom index adds significant value.

The RECom map only shows the benefits of having a wind park at the given location, not the cost of establishing a wind park there. For finding suitable locations, cost drivers such as water depth and distance to shore need to be considered as well. They will to some extend counteract the benefits of the "good" locations identified by the RECom map, which tend to lay far offshore in deeper waters. This is to be adressed as future work.

The focus of this article lays on wind power, but the methodology is not wind-power-specific and it can likely be used for solar power. An adaptation of the RECom index for mixes of different weather-driven renewable energy sources will be considered in future work.

*Author contributions.* Til Kristian Vrana had the idea behind this activity and contributed to developing it and writing this article. Harald G. Svendsen contributed to developing the activity, has realised it in Python and contributed to writing this article.

*Competing interests.* No competing interests are present.

*Acknowledgements.* This activity has been supported by the Ocean Grid project, which is a Green Platform project financed in part by The Research Council of Norway.



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
