# Peer review of "Renewable Energy Complementarity (RECom) Maps – a comprehensive visualisation tool to support spatial diversification"

_Wind Energy Science, 2023_

## Author Comment (AC2)

**RC1**

>> The limitation comments in the conclusions section of the paper contain many of the remarks that I was going to make about the paper. It would be helpful to the reader if these various limitations were mentioned at appropriate points in the main body text, to avoid the reader questioning (for most of the paper) whether the authors were aware of, or had considered, these limiting issues themselves.

**Thank you for this remark. We repeated these considerations where they become relevant in the text to solve the problem you pointed out.**

>> Among the limitations mentioned, should distance to shore not be mentioned, and hence the cost of the supporting infrastructure, and perhaps the possibility of an offshore grid? (I'm not suggesting that additional analysis is completed, but instead the authors shouldn't hide the limitations of the work at the very end of the paper. Address the limitations upfront – they improve the paper.)

Again, thank you for this constructive comment. At the moment, our approach solely addresses the expected revenues from a given site, but ignores the cost of developing that site. For this, distance to shore and water depth, would be the main parameters. We have played with the thought of creating the ultimate "wind power site" map, that covers both costs and revenues, and this could be pursued in future work. These considerations and limitations have been added to the introduction.

>> The authors could also probably note that the potential for contracts for difference, offshore energy hubs (with some storage capability), use of the offshore wind energy to create green hydrogen (or other) rather than connecting in to an electrical grid, etc. All of the above would affect the relationship with electricity price and its importance.

Some of these aspects have been added to the introduction.

**I>> Why are offshore datapoints, Figure 2, based on a rectangular rather than square grid?**

A square grid on a curved spherical surface does not exist. The centre points of the grid are 1 degree latitude and 1.5 degrees longitude apart. The longer longitude separation is because longitude lines are closer farther north. The grid boxes are nearly square at 50 degree latitude, as 1.5 degree in longitude at 50 degrees latitude is 111 km \* cos(50 degrees) = 107 km, i.e. nearly the same. The fact that the data boxes are non-square and of different sizes does not affect the results in any significant way. A comment has been added in the text to explain the rationale for using this lattice.

>> The paper creates a number of colourful figures, which is good, but it is then left to the reader to judge the variations between figures. Would a quantitative metric be useful here to emphasise any improvements which the authors are claiming?

The claimed improvements are more about what important effects the indices plotted in the figures manage to capture, than about reducing some well-defined error. Quantifying the improvement is therefore difficult. It's certainly a good thought, but we are at loss to what metric could be useful for this.

>> When attempting to recognise the impact of price on the value of offshore wind, a few approaches are presented, which, as the authors note themselves, are not entirely satisfactory. The authors seem to have decided that they can only use wind speed time series data to create all the heat maps, but they don't actually declare or justify this assumption. Obviously, electricity price is "influenced" by electrical demand time series (or net demand time series), or some suitable proxy, but it is not explained why such additional data has not been used here.

We agree that there are several important factors that influence the electricity market value, of which solar power output is likely to be the most important factor. Including the influence of solar power on the wind power market value (through correlation between wind and solar resource) is considered for a future expansion of the RECom map, as stated in the text. Adding the influence of correlation with the demand time series could be another interesting future step.

In general, adding more factors may give a more realistic estimation of the market value but at the cost of introducing more assumptions and obscuring the main message of the article. And however many factors are included, we will never be able to fully compute market value because of the inherent uncertainties. We have chosen to focus on offshore wind only to keep the message clear.

We have added a comment on this in the introduction.

>> Page 21 – it is noted that changing the value of beta has a major impact on the heat maps, but it is not made clear how to determine a suitable value for beta, particularly given that one of the advantages of the proposed method is that it is meant to avoid the need for detailed energy system analysis. It seems rather convenient that the authors are deciding that a difficult question to answer is considered out of scope.

The main purpose of the mapping approach introduced in the article is to be a *tool for visualisation* of the dependency between offshore wind capacity deployment (figure 3), wind-price relationship (beta) and the resulting expected value of offshore wind at different locations. It is not an analysis of future capture prices. The beta is to be thought of as an input parameter that can be selected to represent different scenarios for the future power price market, in a similar way as the wind power capacity distribution represents a future scenario.

No analysis would give a definitive answer to what the "correct" value for beta would be, as it is highly dependent on how the energy system evolves in the coming years/decades and therefore very uncertain. It is indeed da difficult question, but mainly in the sense that it is difficult to establish the information needed to compute it. We therefore believe it is an appropriate and transparent choice to leave the value of the beta parameter as an open parameter. The idea is that the approach can be repeated with different assumptions and underlying wind resource datasets.

Comments have been added to the text in the introduction and in Section 5.2.

>> Clearly, spatial diversification is good from a power system perspective, but is there an element of the "prisoner's dilemma" here, whereby unless the individual investor incentives align with the overall power system (electricity market) objectives, then will "short-sighted" investment decisions persist?

Spatial diversification can be driven from the government side by selecting the areas for wind power deployments in a strategic way. Investors will always be short-sighted and looking at their investments. But still, it all also depends on the subsidy schemes for wind power. But the more wind power is exposed to the electricity market, the more beneficial it will be for individual investors to diversify. The topic has been added to the introduction.

>> The formatting for most of the references in the paper is incorrect when using the Harvard style. The complete reference should be included in brackets if the reference doesn't form an "active" part of the sentence, e.g. offshore wind farms are located in the sea (Vrana and Svendsen, 2023).

Citations have been updated.

>> Line 286 no -> do

We have addressed this issue.

>> The paper makes reference to chapters instead of sections

We have addressed this issue.

**RC2**

General response: Most of the comments of reviewer 2 are concerned with the level of accuracy in the modelling of the relationship between wind power output and power price. The comments are insightful and true in many ways, but it is important to keep in mind what the main point of the article is: It is about providing a general method for visualisation without relying on market modelling. To be useful as a that, it needs to rely on as few assumptions and input parameters as possible. The approach is certainly not an alternative to detailed energy system modelling.

Text has been added to the introduction to clarify this point.

>> While its easy to agree that it is not sufficient to use capacity factor maps as the sole source of information where to build new wind power plants, the proposed approach is rather complicated - both to understand how it RECom index is calculated and what is its meaning. This makes the usefulness of the index questionable even though it incorporates the market value aspect in a simplified manner.

We are fully aware of the fact that the presented approach is more complicated than just using capacity factor. However, we have kept it as simple as possible while still achieving the target: including the market value impact. Based on our judgement, we succeeded in only adding complexity where needed. If someone can achieve the set target with a significantly simpler approach, we would be more than interested to hear about it. To our knowledge, no such superior approach exists to date.

>> Furthermore, it can be argued that the index captures the potential market value poorly and unreliably. There is no evidence in the paper that the function depicting the relation between wind power output and electricity market revenue market value factor is realistic. In fact, when looking at market price results from energy system models with high shares of variable renewables, the price duration curves are not like the smooth curve depicted in the manuscript.

We do not claim that the actual relationship has a simple linear form. But it is a fair firstorder approximation: The chosen linear function with negative slope resembles the first order Taylor expansion of any function that expresses a negative correlation between total wind power output and wind power price. Since this negative correlation is accepted as fact by the scientific community, any function depicting a "realistic relation", will have the first order Taylor expansion as selected by us. Higher order expansions or other more complex functions may give a better description of the relationship. But in order to limit the complexity of the approach (as commented above) the utilisation of more advanced functions was discarded. It is impossible to have a "non-complicated" approach and at the same time try to depict the power-price relation in a precise and non-linear manner.

>> It would be better to use market price time series from energy system models directly to calculate market value of infitesimal wind power plant in all the feasible locations. This bypasses all the complexity presented in the method and gives an unambigious indicator based on money. The downside is that one needs to run energy system models or to use time series from existing model runs. Nowadays there are more and more those available in the public domain. As a consequence, the real-life value of the RECom index can be questioned.

The results from an energy system model are highly dependent on the assumptions used, and to take the "right" decisions on what to assume is really not a trivial task. We are fully aware of the energy system model based approach you explained, but the entire intention of our work is to provide an "as-simple-as-feasible" method to consider market value WITHOUT needing energy system models. This might not seem an appealing approach for all who work with energy system modelling, but we believe that such simplified approach still is valuable for many other people.

We aimed at being more accurate than "capacity factor only" considerations, and simpler than energy system model considerations. We are fully aware of that our approach is more complicated than the first, and less accurate than the second. It still appears valuable to us, to establish an intermediate level between to two extremes.

>> On a less fundamental level: while the manuscript describes the methodology relatively clearly (given the complexity of the method itself), there are data-side assumptions that make the provided example questionable. Using Renewable Ninja approach with its old power curve is not up-to-date. Power curves have evolved in the last ten years and there are better available methods to consider the move from single wind turbine power curve to wind power plant level power curve. It could be that this choice does not affect results much, but there is no evidence provided for that in the manuscript.

We are aware of this shortcoming, and the conversion from wind speed to wind power for this type of analyses is something that is being addressed in other work. We believe that the introduced error should be limited, as it appears equally for all wind power plants, leading to a limited influence on the values that are relative to the average. However, the assumed power curve does affect high wind sites and low wind sites differently, introducing an error component that does not disappear for values relative to the average. Analyses to provide evidence for precisely how big this effect is would require detailed wind farm analysis and is therefore not something that can be included in the present paper.

Other simplifications are also relevant in this context, such as the use of wind speed at height 100 m, the implicit wind shear assumption, the low geographical resolution (figure 2) and the omission of wake losses. These are also questionable assumptions. Together with the power curve assumptions, these introduce a kind of bias in the calculations that means the computed capacity factors are not very reliable.

**A comment has been added to the end of Section 2.2**

>> I don't see how the approach "circumvents the dependency on the wind power curve choice". Division by a constant (mean capacity factor) does not change the fact that it would be better to consider wind turbine choice as a function of the wind conditions on a particular site.

Considering different wind turbine types (and wind turbine spacing) depending on the wind conditions would surely add accuracy to the approach, but it would at the same time add significantly to the complexity.

It would be interesting to discuss if such an approach would have more benefits than costs to the approach, to determine if it should be included in a future revision. This mainly depends on if it is possible to include such considerations in a simplified manner, without adding discrete steps (e.g. n discrete wind turbine types with individual power curves to choose from). It might also be problematic to include in a revenue-only consideration, as increasing turbine spacing is a cost benefit consideration, while costs are not accounted for in this approach.

The topic has been addressed in the relating section and in the future work part of the conclusion.

>> In fact the "relative capacity factor" is just a distraction from the actual capacity factor (assuming the power curve that was chosen).

We think that it is not "just" a distraction, as it manages to correct to some level for various biases introduced in the computation, not only from the choice of power curve but also other factors as mentioned above. It enables the comparison of various capacity factor maps that might have been plotted with different individual power curves. Capacity factor maps may change a lot depending on assumptions whereas relative capacity factor maps change significantly less, adding robustness to the approach. The relative capacity factor also needs to be introduced in order normalise at a value of one, for the later inclusion of the market value impact. This explanation has been added to the manuscript.